# Choosing fit-for-purpose biodiversity impact indicators for agriculture in the Brazilian Cerrado ecoregion

Gabriela Rabeschini[1,2] ✉, U. Martin Persson [3] ✉, Chris West [4] & Thomas Kastner [1]

Understanding and acting on biodiversity loss requires robust tools linking biodiversity impacts to land use change, the biggest threat to terrestrial biodiversity. Here we estimate agriculture's impact on the Brazilian Cerrado's biodiversity using three approaches—countryside Species-Area Relationship, Species Threat Abatement and Restoration and Species Habitat Index. By using same input data, we show how indicator scope and design affects impact assessments and resulting decision-support. All indicators show agriculture expansion's increasing pressure on biodiversity. Results suggest that metrics are complementary, providing distinctly different insight into biodiversity change drivers and impacts. Meaningful applications of biodiversity indicators therefore require compatibility between focal questions and indicator choice regarding temporal, spatial, and ecological perspectives on impact and drivers. Backward-looking analyses focused on historical land use change and accountability are best served by the countryside-Species Area Relationship and the Species Habitat Index. Forward-looking analyses of impact risk hotspots and global extinctions mitigation are best served by the Species Threat Abatement and Restoration.

Transformation of the Earth's systems by humans has caused biodiversity to steeply decline in recent history, primarily driven by natural habitat loss or deterioration due to land use (LU) change, with agriculture as its most prominent driver[1]. In order to develop evidence-based actions for achieving the targets of the United Nations' Kumming-Montreal Global Biodiversity Framework, we need robust tools for assessing LU change impacts on biodiversity that provide decision-support for accountability and effective conservation measures.

Biodiversity indicators are tools to monitor ecosystems current conditions, in order to guide governance for safeguarding biological diversity from the local (e.g., on-the-ground implementation in conservation actions) to the national (e.g., national biodiversity strategies and action plans) or global scales (e.g., the Convention on Biological Diversity (CBD)). They can be used to assess historical contributions to biodiversity decline, its current status, and future trends under agricultural land transformation[2,3]. The sustainability of a production system can be evaluated according to its interactions with biodiversity, to estimate how much native biodiversity has been, or will be, lost—or alternatively can be conserved or restored—depending on management strategies. In these contexts, indicators are used to assess (biodiversity) impacts linked to different LU change drivers, which includes accountability for historical impacts or identification of current pressures[4–6]. Such biodiversity information is relevant not only to decisions on agricultural LU locally, but also those concerning supply-chain decisions downstream[7,8].

Whether used by government agencies, research institutions, non-governmental organisations, businesses or consumers, relating

[1]Senckenberg Biodiversity and Climate Research Centre, Frankfurt am Main, Germany. [2]Faculty of Biological Sciences, Goethe University Frankfurt, Frankfurt am Main, Germany. [3]Department of Space, Earth and Environment, Chalmers University of Technology, Gothenburg, Sweden. [4]Stockholm Environment Institute York, University of York, York, UK. ✉e-mail: gabi.rabeschini@gmail.com; martin.persson@chalmers.se

biodiversity indicators to the agricultural LU drivers will be useful if it equips actors with robust information to address trade-offs between production, consumption and conservation. Policy makers may wish for an indicator that is easy to implement and communicate; businesses, for an easy-to-track, target oriented indicator; researchers, for an indicator that captures accurately as many aspects of biodiversity as possible. In practice, however, biodiversity's multidimensional nature imposes a conceptual challenge that cannot be appropriately reflected by one single 'apex' indicator[9] and capturing its dimensions requires alternative metrics. Nonetheless, users might select one main indicator for the sake of simplicity. Given that different indicators emphasise different aspects and, thus, may provide contradictory evidence[10], actors may be tempted to choose indicators that best serve their interests instead of the best fit-for-purpose.

Different perspectives on biodiversity and its conservation—along with data availability and processing constraints—dictate how indicators are designed and, consequently, influence which indicator is best fit for each case and how to interpret its results[11]. For instance, a species conservation focus weights rare and endangered species higher whilst placing less emphasis on abundant species, while an ecological resilience focus might emphasise more abundant or key species that are important for ecosystem function[10]. Similar issues are the baseline choice (i.e. "pristine" or "cutoff") or the extinction risk scale (i.e. local, regional or global). The perspective favoured will depend on what aspects of the impact of agriculture-linked systems on biodiversity users want to emphasise.

In recent years, considerable effort has been made to provide end-users with guidance on the suitability of biodiversity impact indicators[12–14]. Previous comparisons between indicators, however, have not used the same input data, making it difficult for users to understand why results differ and increasing the risk of drawing misleading conclusions or applying the wrong indicators. Importantly, the lack of standardised inputs obfuscates the influence of data uncertainties vs indicator choice and design. In contrast, here we use the same input data—taxa coverage, geographic extent, LU configuration—to assess how the scopes of different indicators affect results and in which ways they are useful for providing decision-makers with information on the role of agriculture as a biodiversity loss driver along supply chains. For this, we use three prominent approaches based on species richness—the countryside Species-Area Relationship (cSAR) model[15,16], the Species Threat Abatement and Restoration (STAR) metric[17], and the Species Habitat Index (SHI)[18,19] (Table 1)—to estimate agriculture's impact on the biodiversity of terrestrial vertebrates in the Brazilian Cerrado.

The Cerrado is the second largest ecoregion in South America, covering around 2 million km², and the world's most biodiverse savanna, holding 5% of global animal and plant biodiversity, including many endemic species[20,21]. However, half of its area has been converted to agricultural LU (Supplementary Fig. 1), accounting for 62% of Brazil's cotton, orange, sugar cane, maize, soybeans, beans, potato, coffee and eucalyptus production and 40% of the country's heads of cattle[22]. In comparison to the neighbouring Amazon ecoregion, the Cerrado has weaker habitat protection laws and enforcement, and areas under deforestation alerts have increased by 43% between 2022 and 2023[23].

To apply the three approaches, we produced three 5km-resolution rasters of the Area of Habitat (AOH) of 2185 native terrestrial vertebrate species found in the Cerrado, for contemporary (2021) and recent (1985) LU patterns, as well as for pristine conditions (i.e. assuming LU absence). We used spatial information on species distribution ranges and habitat preferences from IUCN, land use and land cover (LULC) maps from Mapbiomas (Collection 7.0) and digital elevation models from Open DEM to produce the rasters. We considered agricultural LU all classes with specific crops, cropland, pasture (or a mix of the latter two), as well as monocultural tree plantations (Supplementary Table 1). LUs related to extractivism or mixed uses within semi-natural land cover were not included in this collection of LULC maps. We assessed global impact—how agriculture in the Cerrado contributes to the species' global extinction risk—with the STAR approach and the global-weighted applications of cSAR and SHI. Here, impact is weighted by threatened endemic richness, i.e., the species' threat level following IUCN Red List times the fraction of its range within the region[24]. The regional impact —how agricultural LU in a specific region contributes to the risk of species disappearing from it— was assessed at two levels: at the whole Cerrado ecoregion and in 48 mesoregions, i.e. regions in a geographic area with socioeconomic similarities, across the Cerrado's extent (Supplementary Fig. 1). For this, we used the cSAR and SHI approaches. Finally, we used the cSAR to assess local impacts—risk of biodiversity loss at 5 km pixel-resolution.

In this work, we present and compare these assessments in terms of: (1) total and taxon-specific biodiversity impacts; (2) geographic distribution of biodiversity impacts; and (3) attribution of biodiversity impacts to LU types. We show that, although all metrics point out the leading role of agricultural LU in the increasing pressure on the Cerrado's terrestrial biodiversity, they provide distinctly different insights into biodiversity change drivers and impacts. Unlike the consistent results across spatial scales in detecting biodiversity impacts in traditional agricultural areas, local and regional assessments are more effective than global ones in agricultural frontiers. Overall, the cSAR and SHI approaches are best fit for 'backward-looking' analyses with focus on historical LU change and accountability, while the STAR approach best informs 'forward-looking' analyses focused on

**Table 1 | Description of the biodiversity metrics used in this study**

| Indicators | cSAR[15,16] | STAR[17] | SHI[18,19] |
|---|---|---|---|
| Output | Potential species loss (no. of species committed to extinction) | Threat abatement score (STAR$_T$ score) | Percentage of habitat area size and connectivity lost |
| Estimation | Estimates species loss per unit of land-use change in an area, and attributes it to land use types that replace natural habitat | Estimates how much abating a threat in the species' remaining habitats contributes to reducing the species' global extinction risks | Estimates changes in the habitat area size and habitat connectivity of a species or an assemblage of species in an area |
| Data requirements | Species' AOH maps; Baseline species richness; Area of LULC types (present and historic);Species affinity to each LULC type; Species' extinction risks classification | Species' AOH maps; Species' extinction risks classification; Classification of threats that cause population decline | Species' AOH maps; Size of species' habitat area; Connectivity of species' habitat area; Species' extinction risks classification |
| Spatial scale of impact on species | Local; Regional; Global | Global | Regional; Global |
| Temporal scale of impact on species | Pristine to 2021; 1985 to 2021 | Present (currently threatened species) | Pristine to 2021 1985 to 2021 |

cSAR countryside Species-Area Relationship, STAR Species Threat Abatement and Restoration, SHI Species Habitat Index, AOH Area of Habitat, LULC Land use and land cover.

**Table 2 | Impacts of agriculture per taxon.** Impacts of agriculture on the terrestrial vertebrate species of the Cerrado ecoregion by 2021 measured by the countryside Species-Area Relationship (cSAR) model, the Species Threat Abatement and Restoration (STAR) metric and the Species Habitat Index (SHI)

| | | Unit | Amphibians | Reptiles | Birds | Mammals | Total |
|---|---|---|---|---|---|---|---|
| cSAR | Regional potential loss | Number of species | 37 | 55 | 147 | 48 | 287 |
| | Global-weighted potential loss | Number of species | 1.7 | 3.8 | 5.9 | 2.3 | 14 |
| STAR | $STAR_T$ score for agricultural threats | - | 2758 | 4619 | 5390 | 2630 | 15,397 |
| | Agriculture's share in total $STAR_T$ score | % | 58 | 72 | 58 | 51 | 60 |
| SHI | Regional habitat loss | % | 25 | 35 | 37 | 38 | 35 |
| | Global-weighted habitat loss | % | 28 | 32 | 39 | 35 | 35 |

Source Data is provided in ref. [44].

mitigation of global extinctions or identification of impact risk hotspots. Our standardised setup allows us to discuss how and where there is potential for granular and complementary biodiversity metrics to jointly inform landscape-level attributions of biodiversity risk to agri-production and supply chain activities, and in turn contribute to decision-making processes and appropriate conservation responses.

## Results
### Total and taxon-specific biodiversity impacts
All three metrics indicate agricultural LU as the major cause of biodiversity decline in the Cerrado, though the relative importance of agriculture differs between indicators. In cSAR, approximately 98% of the potential species loss by 2021 is associated with agricultural LU at all impact scales assessed. At ecoregion level, the potential regional loss is 287 species, representing around 13% of the species found in the ecoregion (Table 2). The global-weighted impact estimates 14 potential global species losses, varying between 12-14% across taxonomic groups (Table 2).

In STAR, agricultural threats are expected to cause population decline for 470 species, which corresponds to 3/4 of the species experiencing threats in the Cerrado. In this approach, each species has a global threat abatement ($STAR_T$) score over its total range, according to the threats assigned to it. If all threats would be abated, the $STAR_T$ score would be 0. On average, agricultural threats account for 62% of the species' total $STAR_T$ score, though for 81 species, agriculture is the only threat. That the STAR metric includes other threat categories—e.g. pollution, or invasive alien species—explains the reduced relative importance of agriculture when compared to the other metrics which focus solely on LU. Agricultural threats have a bigger share in the reptiles' total $STAR_T$ score than in those of the other taxa (Table 2).

In SHI, 99% of the 1478 species that cannot inhabit agricultural LUs had a decrease in their habitat ecological integrity—i.e. habitat area size and connectivity—by 2021. At ecoregion level, the SHI estimates an average loss of 35% in the species' habitat ecological integrity within the Cerrado compared to pristine conditions. The average loss in the global-weighted SHI remains the same, with small variations across taxa. Amphibians have a smaller loss in both impact scales compared to other taxa (Table 2).

Exploring temporal trends on impact can also be relevant. With cSAR and SHI, impacts can be compared across different years as long as there is LU information. For instance, agriculture's impact at ecoregion level by 1985 was estimated with cSAR as 181 potential regional species losses, or a global-weighted loss of 9 species, indicating that the losses in the Cerrado by 2021 are 58% larger than they were by 1985. Using SHI, the species that cannot inhabit agricultural LUs had a 16% loss in their habitat ecological integrity by 1985, implying that more than half of the loss found by 2021 happened in this 36-years window. In STAR, the score calculation is based on the current threats to species following IUCN's Threat Classification Scheme, and, thus, such a temporal comparison is not within the metric's scope.

### Geographical distribution of biodiversity impacts
At global level, the geographical distribution of impacts assessed with the STAR metric diverges more greatly from the ones assessed with the other two indicators. cSAR and SHI also agree considerably at regional level, with the exception of some areas in the northeast Cerrado (Fig. 1).

In terms of global impacts, the mesoregion with the highest number of potential global species losses, as assessed with both cSAR (1.8 species) and SHI (0.36% of loss in habitat ecological integrity), is South Goiás (mesoregion 47, see Supplementary Fig. 1) (Fig. 1a, b). A fraction-of-a-species loss can be interpreted in this context as parts of a species' population being lost in a region[25]. In STAR, the mesoregions with the highest $STAR_T$ scores, varying from 2706 to 2220, are spread across the central Cerrado, from west to east (Fig. 1c). In contrast to the other two metrics based on lost habitat, the $STAR_T$ score will demarcate areas where more threatened and endemic species have more remaining habitat.

When it comes to the regional impacts assessed with cSAR, the mesoregions with the highest potential regional species loss, varying from 312 to 256 species, are mostly in southern Cerrado (Fig. 1d). In SHI, the greatest decrease in habitat ecological integrity, varying from 41 to 51% loss, are mostly in southwest Cerrado (Fig. 1e). Interestingly, the biodiversity impact on the MATOPIBA region—an agricultural frontier between the states of Maranhão (MA), Tocantins (TO), Piauí (PI) and Bahia (BA)[26]—shows in the SHI approach, but not in the cSAR.

Finally, at the local scale, the cSAR metric shows higher potential local species loss in the southern Cerrado, similar to the global and regional assessments (Fig. 1a, d), but it also points out particular locations with high impact that might have been 'masked' at regional or global scales by the considerable size of the surrounding remaining natural habitat (Fig. 2a). This is especially demonstrated by the high scoring pixels in the upper/central east and west parts of the region. Global level assessments may be less sensitive at detecting impacts on areas in current agricultural frontiers because of the mixed attributes of heavily converted local patches and a dense area of species' natural habitat in the landscape. It is also worth noting that areas with higher potential local species loss in the cSAR are those with lower scores as shown by the global $STAR_T$ score disaggregated to pixels (Fig. 2a vs Fig. 2b). This contrast between metrics occurs because the $STAR_T$ score is focused on species' current AOH and does not account threats' historical impacts. This has the practical implication that areas with intense historical LU change, where species have little or no habitat left, will have low $STAR_T$ scores (i.e. there is no current threat in an area where the species' habitat was already converted), but high impacts when assessed with cSAR and SHI, as is evident in Fig. 2. Here, it is important to bear in mind that, although spatially disaggregated to pixels, the $STAR_T$ score reports global impacts.

### Attribution of biodiversity impacts to land use types
For the attribution to LU types, we first compare the assessments of the global-weighted impact at the level of the whole Cerrado

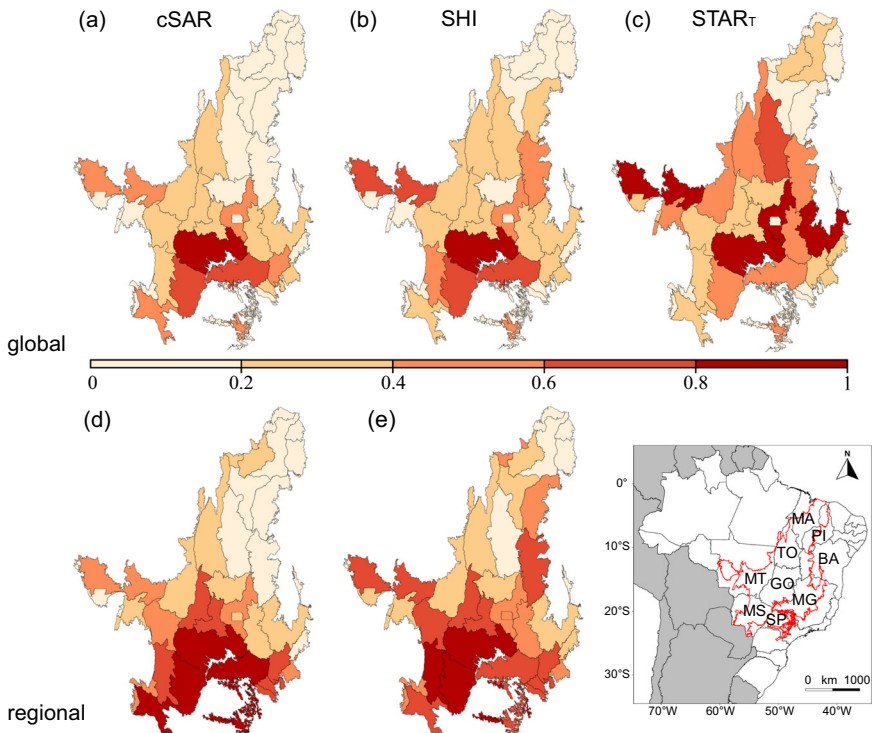

**Fig. 1 | Indicators comparison at different extinction scales.** Potential global (**a**) and regional (**d**) loss of native vertebrate species in Cerrado mesoregions by 2021 assessed with the countryside Species-Area Relationship (cSAR). Global (**b**) and regional (**e**) loss of habitat integrity in Cerrado mesoregions by 2021 for the native vertebrate species that cannot inhabit agricultural land uses assessed with the Species Habitat Index (SHI). Global threat abatement score (STAR_T) disaggregated at mesoregions (**c**) assessed with the Species Threat Abatement and Restoration metric. Data values were scaled between a range of 0 (lowest) to 1 (highest) with a min-max normalisation for intercomparison between indicators. The Brazilian Cerrado ecoregion is highlighted in red in the map of South America at the bottom right, showing the Brazilian states within its coverage. MA Maranhão, TO Tocantins, PI Piauí, BA Bahia, MT Mato Grosso, GO Goiás, MS Mato Grosso do Sul, MG Minas Gerais, SP São Paulo. Source Data is provided in ref. 44.

ecoregion. In cSAR, pasture is the LU type with the greatest potential global species loss (7 species), followed by soy (3 species), mosaic of agriculture and pasture (i.e. areas where the remote sensing data could not distinguish between pasture and agriculture, 2 species), tree plantation (0.5 fraction of a species) and sugar cane (0.4 fraction of a species).

In STAR, allocation is based on the threat categories of the IUCN's Threat Classification Scheme[27]. The three categories of agricultural threats—Annual & perennial non-timber crops, Livestock farming and ranching, and Wood & pulp plantation—can be disaggregated to farming scale, e.g. small-holder or agro-industry, but not to individual LU types. Annual & perennial non-timber crops had the highest STAR_T score, 8785, followed by Livestock farming & ranching with 5095, and Wood & pulp plantation with 1518.

It is worth noting that differing criteria for allocations result in different patterns. For instance, in cSAR, more impact is allocated to pasture based on the greater area shares of this LU and the lower species affinities to it, whereas in STAR, more allocation is given to Annual & perennial non-timber crops because more species are classified as currently threatened by it in the IUCN's Threat Classification Scheme.

In the global-weighted cSAR at the mesoregion level, pasture is the LU type with highest potential global species loss in 31 mesoregions, followed by soy in 7 mesoregions, such as north Mato Grosso (mesoregion 38) and the MATOPIBA area (mesoregions 9, 12 and 14), which evidences the biodiversity impacts linked to soy production in these new agricultural frontiers (Fig. 3). In STAR, when the global STAR_T score is disaggregated to mesoregions, all 48 mesoregions have higher scores for Annual & perennial non-timber crops than for the other two agricultural categories. When the score is disaggregated to

pixels, there are patches with higher scores widespread through the Cerrado for Annual & perennial non-timber crops, while there is a concentration of higher scores in southeast regions for Livestock farming & ranching (Fig. 4).

One may also be interested in how the impact of specific LU types on biodiversity has changed throughout time. Comparing the local impact of pasture and soy by 1985 and by 2021 as assessed with cSAR, shows an overall increase of the biodiversity impacts of both LUs in the Cerrado (Fig. 5). Interestingly, the decrease in the local impacts of pasture in parts of the south can be related to a substitution of pasture by soy cultivation, as there is a reciprocal increase in the impacts caused by soy in most of these areas. This illustrates that cSAR's attribution to specific LUs is dependent on the LU composition used as a comparison to the pristine baseline, disregarding historical transformation between LUs. This is especially important in approaches for attributing occupational LU impacts, like in standard Life Cycle Assessments (LCAs), as, particularly in areas with large-scale land-dynamics like those in central and southern Cerrado, not accounting for LU substitutions can have potentially large effects on estimated characterization factors. To allocate impacts over LUs that have been present throughout time, the cSAR model could be run year by year with annual LU maps, to calculate a compound score averaged over the years.

The standard application of the SHI approach does not attribute loss of habitat ecological integrity to specific LU types. For such attribution to be possible, the changes in each LU type must be tracked in the LULC maps and then proportionally allocated to the losses in habitat area size and connectivity driven by the LU transformation. By building on the original calculations for the area size component of the SHI, we explored a complementary way to use the SHI approach to

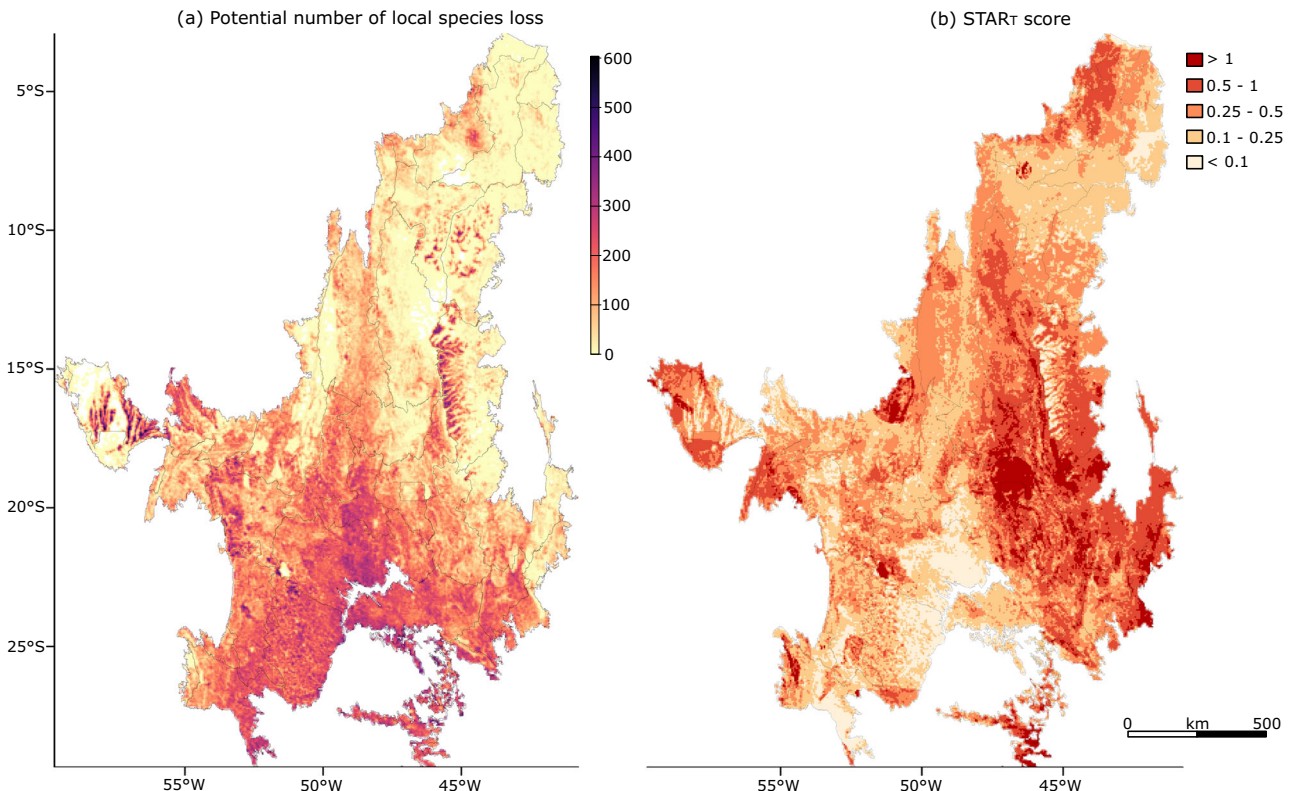

**Fig. 2 | Impacts at pixel-resolution. a** Potential local loss of terrestrial vertebrate species richness due to agricultural land use in the Cerrado ecoregion by 2021 at 5 km pixel resolution assessed with the countryside Species Area-Relationship model. **b** Global threat abatement score (STAR$_T$) disaggregated at 5 km pixel resolution. Source Data is provided in ref. 44.

attribute loss in habitat area in Cerrado mesoregions to specific LUs. This exercise can be found in Supplementary Fig. 2. The shares of the specific LU types in the total loss that resulted are similar to those found with cSAR (Fig. 3).

## Discussion

Different indicators capture different components of the biodiversity-impact process, which—if properly applied by stakeholders, can provide useful information tailored to different policy-relevant goals—or if not, lead to confusion and poorly supported decisions. Although indicators are not fully incongruent, neither in their conceptualization, nor in their operationalization, the comparison presented above can facilitate decisions on which indicators will be more fit-for-purpose to particular applications.

A meaningful application of the indicators has to consider their temporal and spatial foci. The STAR$_T$ metric is a forward-looking approach, attributing impacts based on current and future threats to biodiversity and focusing on areas of remaining habitat that can potentially be conserved. Thus, within the policy landscape, the STAR metric can be a useful tool for identifying measures to mitigate future biodiversity losses. For instance, only 13% of the Cerrado area is currently under conservation protection[28], considerably short of the CBD's target of protecting 30% of land ecosystems by 2030. The STAR$_T$ mapping of biodiversity risk hotspots can facilitate the identification of areas of conservation importance and shows that abating the agricultural expansion of annual and perennial non-timber crops, especially in the eastern Cerrado and in localized regions of its transition zones with the Atlantic and Amazon forests and the Pantanal wetland, would contribute most to avoiding global biodiversity extinctions. Such measures would be particularly important for threatened reptile species, for which agriculture represents over 70% of their threat score.

The SHI and the cSAR in turn are backward-looking approaches that measure impacts on biodiversity based on LU transformations, emphasizing accountability for what has been lost. Such metrics can be useful tools in biodiversity accounting and disclosure within Corporate Social Responsibility or Environment, Social and Governance reporting[29]. As a prominently used indicator in LCA, it is important to consider that applying the cSAR only to LULC maps with the current LU configuration can disregard potential LU dynamics and give a skewed picture of which LUs (and hence actors) are accountable for historical extinction risks (e.g., disregarding habitat once cleared for use as pasture that more recently is converted to soy production). There can also be an interaction between baseline and time-lags between LU change and biodiversity losses (i.e., extinction debts) in relation to accountability. In a pristine baseline, the emphasis falls on (long-term) accountability, as some of the losses will be impossible to undo (i.e., global extinctions). However, if assessed over shorter historical periods with a more recent baseline, a backward-looking metric such as the SHI may be relevant for mitigatory measures like rehabilitation or reforestation by identifying areas where reversals of recent LU loss can help avoid extinction debts, in particular through reducing habitat fragmentation.

Effective governance for biodiversity in areas such as the Cerrado, where a multitude of social-ecological systems coexist, needs to draw on assessments that best capture the dynamics of their different biophysical components and socioeconomic histories and should contribute to targeted actions for those regions. As an example, the consistently high impacts on biodiversity captured at all scales by cSAR and SHI in south Goiás (mesoregion 47, see Supplementary Fig. 1) is a striking result of historical LU processes in the state, which—due to a mixture of national and international incentives and biophysical characteristics favourable to the technological packages within Brazil's

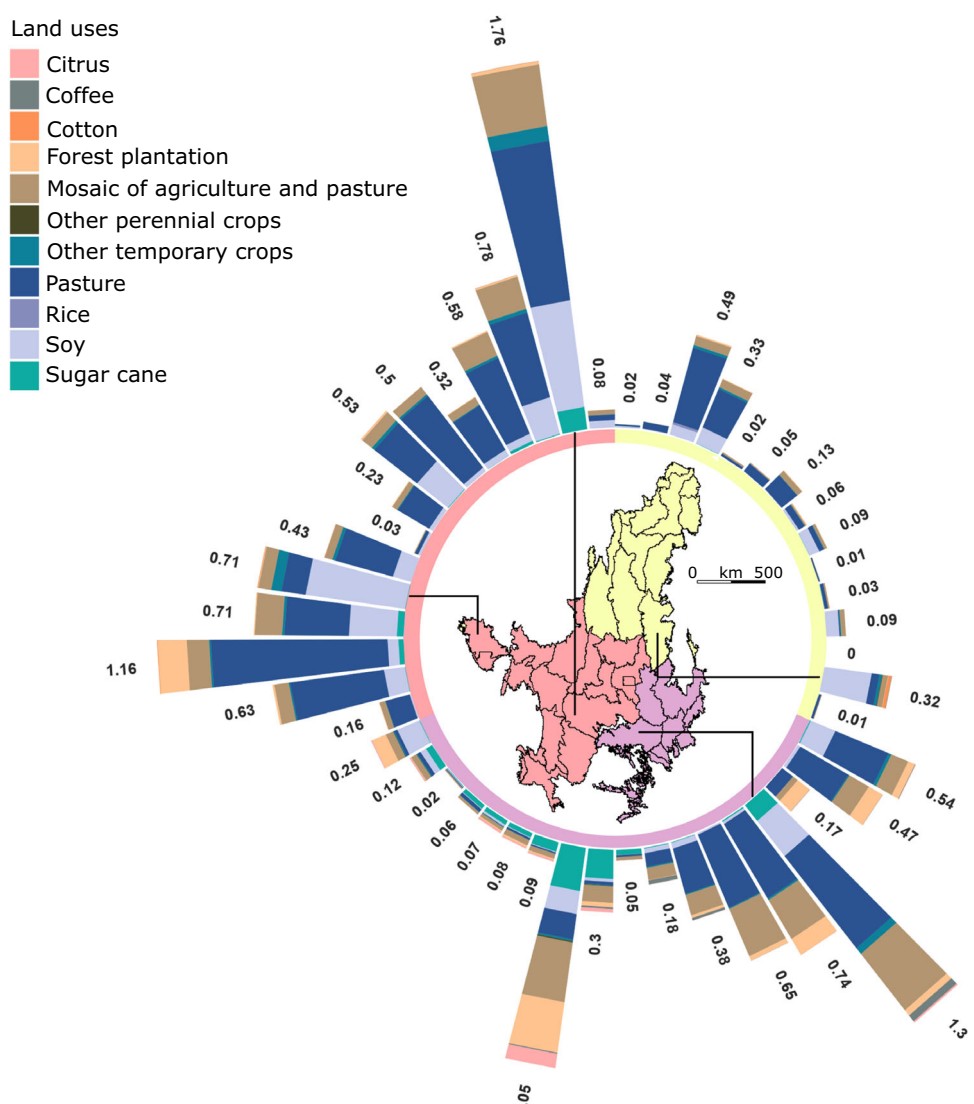

**Fig. 3 | Attribution of impacts to specific land uses.** Number of potential native vertebrate species loss attributed to agricultural land uses on 48 mesoregions of the Cerrado by 2021 assessed by the global-weighted countryside Species-Area Relationship. The loss of a fraction of a species can be interpreted in this context as parts of a species' population being lost in a mesoregion. Mesoregions 1-15, marked in yellow, are in North and Northeast Brazil; mesoregions 16–33, marked in violet, are in Southeast Brazil; and mesoregions 34–48, marked in pink, are in Central-West Brazil. Source Data is provided in ref. 44.

Green Revolution—already had 50% of its extent converted to agriculture by 1986[30,31]. The localised information supplied by the application of such biodiversity indicators can support, for example, the enforcement of the Brazilian Forest Code through spatial planning for restoration focused on priority areas for biodiversity. The Forest Code establishes areas of natural vegetation under permanent protection and legal reserves in rural properties and for which Goiás state had a vegetation deficit of 14,467 km² by 2017[32].

In contrast to the relative levels of consistency across spatial scales observed from the metrics when applied to traditional agricultural areas throughout Cerrado, our results indicate that assessments of local and regional impacts are more sensitive than global level assessments at detecting effects on biodiversity in areas that have been subject to more recent habitat conversion. Such agricultural frontiers, like the MATOPIBA region and northern Mato Grosso (mesoregion 38), are ecological transition zones between the Caatinga and the Amazon biomes, respectively, and face increasing ecological vulnerability and climate pressure due to rapid agribusiness expansion and intensification together with underlying climate change[33,34]. In

such cases, a combination of approaches to detect where local biodiversity has been significantly impacted and where threat abatement efforts would be best enacted to avoid global extinctions, as shown in Fig. 2, would be an informative way for a biodiversity-inclusive spatial planning to mitigate further local deterioration in ecological conditions.

Information on biodiversity impacts is not only required to monitor the proximate drivers of LU change, but it is also essential to identify key stakeholders, link impact to supply-chain decisions downstream and inform both regulatory and voluntary schemes, as well as evaluate strengths and weaknesses of interventions. For instance, according to data from the Trase platform[35], municipalities in east Tocantins (mesoregion 4) traded soy to 16 nations plus the European Union economic bloc in 2020, through 48 trading companies, with China, the European Union and Turkey as the three biggest partners in traded volume and Bunge, Vietnam Agribusiness Limited and LDC Tianjin International Business CO LTD as the three biggest trading companies. This mesoregion has the highest STAR$_T$ score in the Cerrado for the broad LU category Annual & perennial non-timber crops and meaningful

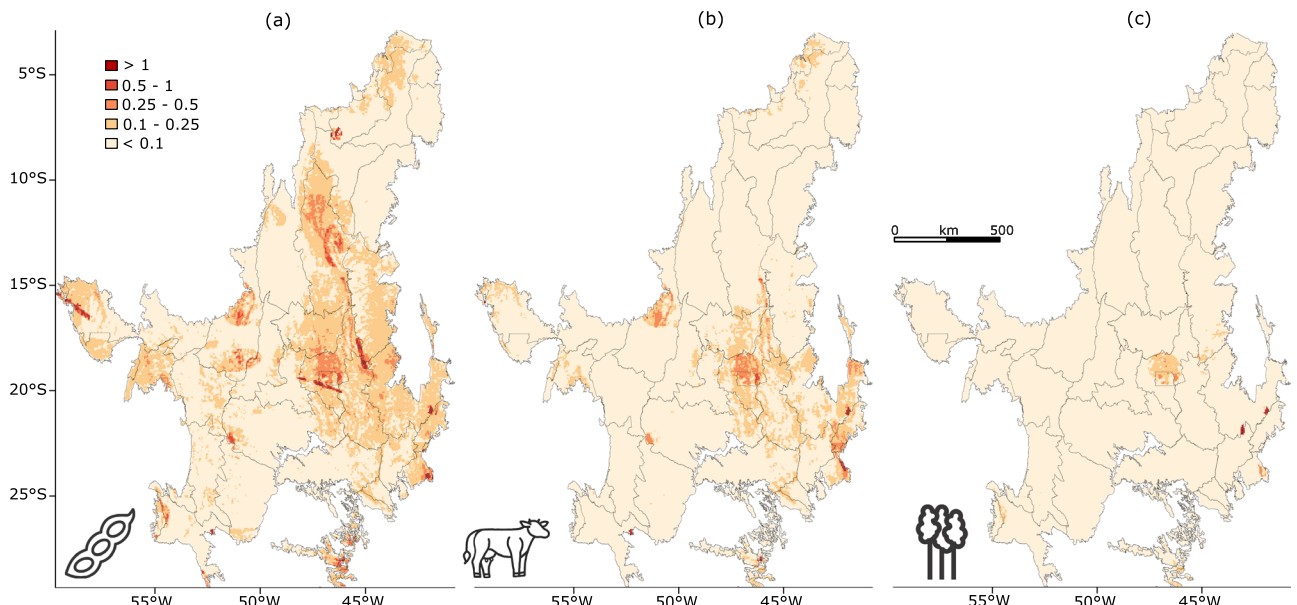

**Fig. 4 | Attribution of impacts to broad agricultural threats.** Global threat abatement score (STAR_T) visualized at 5 km pixel resolution and disaggregated by the broad threat categories (**a**) Annual & perennial non-timber crops, represented by the soybean icon; (**b**) Livestock farming & ranching, represented by the cow icon; and (**c**) Wood & pulp plantations, represented by the trees icon. Broad threat categories follow the IUCN threat classification for species. Source Data is provided in ref. 44.

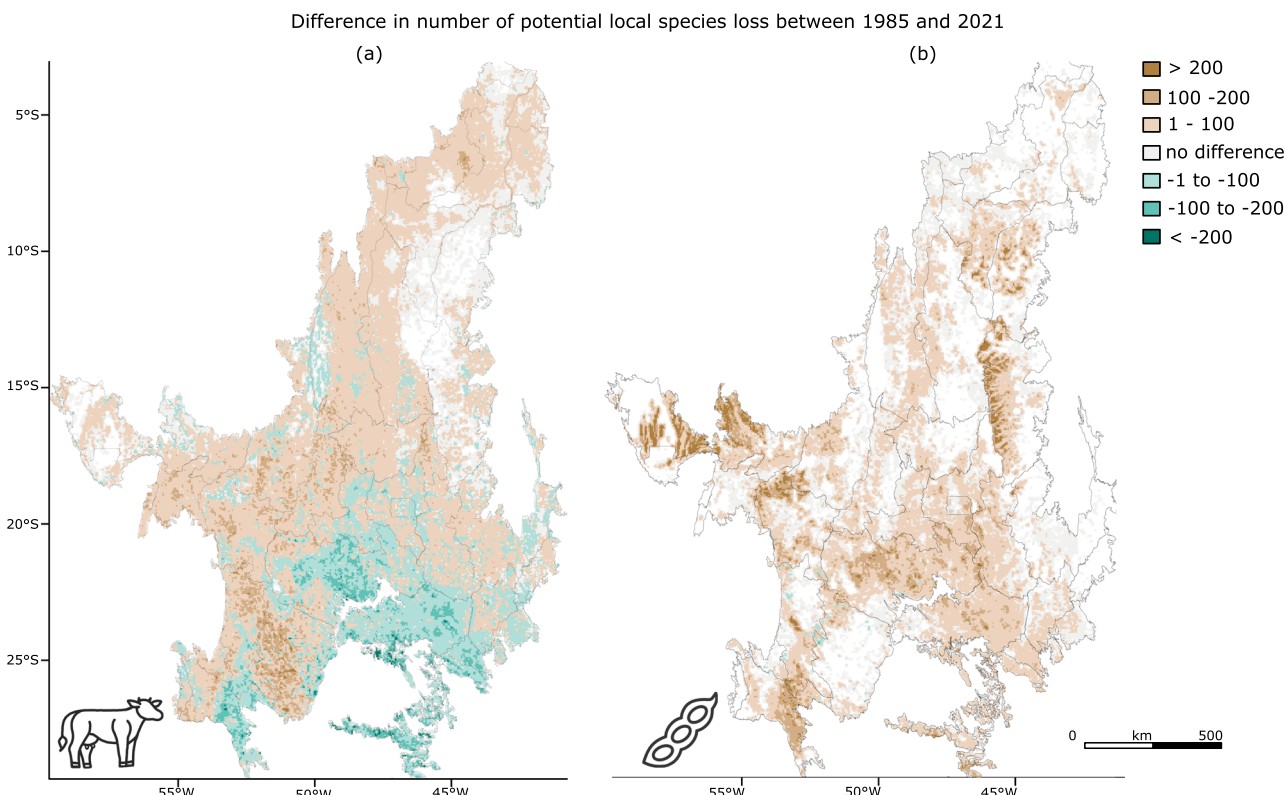

**Fig. 5 | Attribution of impacts through time.** Changes in the potential local species loss attributed to (**a**) pasture land use, represented by the cow icon, and (**b**) soy land use, represented by the soybean icon, between 1985 and 2021 calculated with the countryside Species-Area Relationship, at 5 km resolution. Changes are shown as the differences between the local impacts by 2021 and the local impacts by 1985. Source Data is provided in ref. 44.

efforts to abate this threat should involve dialogue with countries and trading actors downstream in supply chains for planning and, potentially, resourcing, the mitigation of biodiversity risks related to the soy production system, the main cultivated crop in the region.

Attributing impact to specific LU products with measures such as cSAR allows additional finer footprinting by linking the conversion of natural ecosystems to productive outputs and then linking these productive outputs to consumptive demand. Here, calculating a

compound score−different versions of a footprint with different attributions to LU systems that varied over time−is particularly important for regions with a long history of agricultural use, such as southeast Mato Grosso do Sul (mesoregion 37). There, demand-side actors related to pasture systems would have an increased final footprint in comparison to the footprint calculated only based on the current LU configuration, which gives more weight to consumptive demand linked to soy (Fig. 5). Undertaking this sort of exercise can be valuable for understanding the sectors ultimately responsible over varying time periods and, thus, providing more accurate information to interact with those stakeholders.

The analysis conducted here demonstrates the potential for granular and complementary biodiversity metrics to jointly inform landscape-level attributions of biodiversity risk to agri-production and supply chain activities, and in turn their role in informing appropriate conservation responses. The advanced LULC mapping data provided by Mapbiomas for the Cerrado considerably reduces data gaps which would otherwise act as a considerable source of uncertainty for the analysis presented in this paper and−unless there are systematic biases in errors of omission and commission for given LUs or across the landscape−misclassifications are not likely to affect the quantitative results to any great extent. However, if a goal of applying biodiversity metrics of this kind is to attribute biodiversity loss to a suite of commodities (such as might be warranted via LCA activities), uncertainties associated with this type of analysis are likely to get much larger as spatial data on cultivation is only available for selected crops. Additionally, for other parts of the world where ongoing conversion of forests and other natural ecosystems is happening data availability poses a serious issue[36] and is likely to affect the accuracy of biodiversity indicator assessments.

Although some of the differences found across indicators may stem from the biogeographic specifics of the Cerrado ecoregion, most distinctions relate to their conceptual differences and the conclusions from their comparison revealed here are transferable to LUs other than agriculture, as well as to other regions experiencing rapid LU change. To ensure the best fit between the needs of decision-makers, practitioners, and the public and what indicators can and cannot assess, there must be more awareness when establishing the criteria for judging an indicator's suitability, without assuming that a single indicator will give information that is fit for all cases. The cSAR, STAR and SHI approaches are complementary, although with caveats. Good practice towards a more fit-for-purpose use of indicators requires that the focal questions intended to be answered by the assessments and the choice of indicator are matched in terms of the relevant temporal, spatial, and ecological perspectives relevant to understanding impact on biodiversity.

## Methods
### Ethics & Inclusion
This research did not undertake in loco data collection, but has a local researcher in the authors' team. We have also taken local and regional research relevant to our study into account in citations.

### Species and land use information
Species range maps, extinction risk categories, threat classification, and habitat affinity data were obtained for all amphibians, reptiles, birds and mammals in the Cerrado from IUCN Red List (version 2022-2)[27]. We used 30 m-resolution LULC maps from 2021 from the Mapbiomas platform (Collection 7.0), overlaid with digital elevation maps and IUCN species' range maps to produce 5-km-resolution rasters of each species' contemporary Area of Habitat (AOH) within the Cerrado[37,38], excluding parts of the species' range maps where presence was coded as extinct or possibly extinct. We constrained the species range maps to areas within compatible elevation range and habitat type according to IUCN Red List assessments. We excluded exclusively aquatic species and aquatic LULC types from the analyses.

We aggregated the 22 terrestrial LULC types in Mapbiomas into twelve categories to match the IUCN species' habitat classes: Arable Land (including soybeans, rice, other temporary crops, and mosaic of agriculture and pasture); Plantation (including tree plantation, coffee, citrus, other perennial crops, sugar cane, and cotton - complying with IUCN habitat classification description for the plantation category); Pasture; Forest; Mangrove; Savanna; Grassland; Wetland; Natural Non-Vegetated Areas (including rocky outcrop, beach and dune, and salt flat); Urban; and Other Non-Vegetated Areas (including mining, and other non-vegetated areas, like rural infrastructure). Pasture are areas predominantly planted with grasses linked to livestock rearing activity. Natural grasslands used for livestock grazing are predominantly classified as Grassland, which may or may not be grazed. Tree plantations are areas where the natural land cover was converted to areas with tree species planted for commercial purposes (e.g., pine, eucalyptus, araucaria), usually in monoculture.

To determine the species' AOH in the recent past, we followed the same steps described above using the Mapbiomas LULC maps from 1985, this time also including parts where presence was coded as extinct or possibly extinct. To determine the species AOH in the distant (pre-large-scale human activity) past we considered the whole extension of the species' range maps, constrained to areas within compatible elevation ranges. Species past ranges were not extended beyond IUCN's current species distributions range maps due to lack of historical distribution range maps.

### Countryside Species-Area Relationship (cSAR)
The cSAR uses information on the area occupied by different LULC types, baseline richness, and the species affinity to each LULC type to estimate the potential species loss to LU change in a given region, and attribute this to the different LULC types that have replaced natural vegetation. The potential species loss evaluates the number of species committed to extinction due to the LU configuration found in an area in relation to the potential species richness that could be found in the same area in a natural habitat condition. The cSAR approach has mostly been used at large scale resolutions, i.e. biogeographical regions[5,39], but can be applied at finer pixel resolutions, i.e., 10 km[16].

The countryside Species-Area Relationship model[16] estimates the number of potential species loss for each taxonomic group $g$ in location $j$ ($Sloss_{g,j}$) as:

$$Sloss_{g,j} = Spot_{g,j}\left[1 - \left(\frac{\sum_{b=1}^{b_n} h_{g,b,j}A_{b,j}}{Apot_j}\right)^z\right] \qquad (1)$$

where $Spot_{g,j}$ is the potential species richness in original natural habitat conditions; $Apot_j$ is the potential area of natural habitat in location $j$, i.e., the land area in the respective location; $h_{g,b,j}$ is the affinity parameter of taxon $g$ to each broad LULC type $b$ in location $j$; $A_{b,j}$ is the area occupied by each broad LULC type $b$ in location $j$; and $z$ is the SAR exponent for non-forest ecoregions, obtained from ref. 40 and equivalent to 0.23. The SAR exponent indicates how rapidly species are lost in an ecosystem as it loses natural habitat.

The potential species richness in location $j$ ($Spot_{g,j}$) is given by the number of species that have their pristine AOH overlapping the location. The potential species richness was then assessed by creating a count of all species present in the area. The potential area of natural habitat ($Apot_j$) is the entire terrestrial area of location $j$.

The species affiliation to each broad LULC type (natural habitat, arable land, plantation, pasture, urban and other non-vegetated areas) was calculated based on the IUCN Red List Habitat Classification Scheme, which provides species-specific information on habitat preferences. The affinity of taxon $g$ to the broad LULC type $b$ in location $j$ ($h_{g,b,j}$) was given by the number of species affiliated with the broad LULC type $b$ in location $j$ ($S_{g,b,j}$) divided by the number of species

expected in the location under natural habitat cover ($S_{g,j}$), raised to the power of $\frac{1}{z}$ (ref. 41):

$$h_{g,b,j} = \left( \frac{S_{g,b,j}}{S_{g,j}} \right)^{\frac{1}{z}} \qquad (2)$$

For natural habitat land cover, affinity equals 1 as $S_{g,b,j} = S_{g,j}$. The area of natural habitat land cover currently found in the location was given by the sum of the areas classified as Forest, Savanna, Grassland, Wetland, Mangrove, Beach and Dune, Salt Flat and Rocky Outcrop in Mapbiomas.

The total potential species loss can be allocated to each individual LU category based on their area share and the taxon affinity to them. Following ref. 39, the allocation factor of LULC type $b$ in location $j$ ($a_{b,j}$) is:

$$a_{b,j} = \frac{A_{b,j}\left(1 - h_{g,b,j}\right)}{\sum_{b=1}^{N} A_{b,j}\left(1 - h_{g,b,j}\right)} \qquad (3)$$

where $N$ is the number of LULC types in location $j$. The allocation factor is then:

$$Sloss_{g,b,j} = Sloss_{g,j} \times a_{b,j} \qquad (4)$$

where $0 < a_{b,j} < 1$, and $\sum_{b=1}^{N} a_{b,j} = 1$.

Finally, to assess global impact, the impact is weighted by the threatened endemic richness[24]. For this, we first calculate the endemic richness ($ER_{g,j}$) of each taxon $g$ in location $j$ as:

$$ER_{g,j} = \sum_{s=1}^{m} \frac{GR_{s,g,j}}{GR_{s,g}} \qquad (5)$$

where $m$ is the total number of species of taxa $g$ found within location $j$, $GR_{s,g,j}$ is the area of species $s$ habitat range within location $j$, and $GR_{s,g}$ is the total (global) area of species $s$ habitat range. The range fraction of each species $s$ in location $j$ is then multiplied by its threat level ($TL$) according to IUCN Red List to calculate the threatened endemic richness ($TER$) per taxa $g$ in location $j$:

$$TER_{g,j} = \sum_{s=1}^{m} TL_{s,g} ER_{s,g,j} = \sum_{s=1}^{m} \frac{TL_{s,g} GR_{s,g,j}}{GR_{s,g}} \qquad (6)$$

$TL$ is a linear rescaling of the categories defined in IUCN Red List from 0.2 to 1 (least concern, 0.2; near threatened, 0.4; vulnerable, 0.6; endangered, 0.8; and critically endangered, 1).

To calculate the global weighted impact, the potential species richness ($Spot_{g,j}$) in Eq. 1 can be then substituted by the threatened endemic richness ($TER_{g,j}$):

$$Sloss_{g,j} = TER_{g,j} \left[ 1 - \left( \frac{\sum_{b=1}^{b_n} h_{g,b,j} A_{b,j}}{Apot_j} \right)^z \right] \qquad (7)$$

**Species Threat Abatement and Restoration (STAR)**
In the STAR metric, the STAR threat abatement (STAR$_T$) score uses information on species' extinction risks and on threats that can cause population decline to estimate the proportional effect that abating a threat in the species' remaining habitats represents in relation to the global extinction risk imposed by all the threats to this species. All risk categories were included in the analysis, as threats are currently coded for the majority of species on the IUCN Red List. Data Deficient species were excluded. Description of threats include timing (past, ongoing,

future); scope (the percentage of the population affected by the threat); and severity (the rate of population decline caused by the threat within its scope). Threats with past, unlikely to return timing were excluded. Threats with a combination of scope and severity that is not expected to lead to population decline were also excluded (including severity coded as no decline and a combination of severity coded as negligible decline and scope coded as affecting either the minority or majority of the species' distribution, see ref. 17). By doing so, any species assigned to threats that were not expected to result in population decline were not considered in the analysis. Although scope and severity data are mostly complete for birds, this information is still lacking for some amphibian, reptile and mammal species. Ref. 17 explored approaches to deal with missing scope and severity data and concluded that using the intermediate classification of the possible values of scope and severity to replace unknown or missing data was a suitable approach (the intermediate classification for scope is Majority (50-90%), and the intermediate classification for severity is Slow, Significant Declines).

A global STAR$_T$ score was calculated for 635 vertebrate species that are threatened with population decline within the Cerrado ecoregion (147 amphibians, 143 reptiles, 210 birds and 135 mammals), representing 29% of the total species.

The STAR$_T$ score for pixel $n$ and threat $t$ ($T_{t,n}$) is:

$$T_{t,n} = \sum_{s}^{N_s} P_{s,n} W_s C_{s,t} \qquad (8)$$

where $P_{s,n}$ is the percentage of the total current AOH of species $s$ within pixel $n$; $W_s$ is a factor weighted by the risk category of species $s$ according to IUCN Red List assessment (Least Concern = 1; Near Threatened = 2; Vulnerable = 3; Endangered = 4; Critically Endangered = 5; see ref. 17); $C_{s,t}$ is the relative contribution of threat $t$ to the extinction risk of species $s$; and $N_s$ is the total number of species in pixel $n$.

The relative contribution of a threat to the total extinction risk of a species is the percentage of population decline expected to be caused by that threat, reproduced from ref. 17 (see Supplementary Table 2), divided by the sum of the percentage population declines of all threats affecting this species. For instance, if a species has three threats, T1, T2 and T3, expected to cause a population decline of 18%, 9% and 5%, respectively, the relative contribution of threat T1 to the total extinction risk of the species will be $18 / (18 + 9 + 5) = 0.56$. The effect of the weighting factor for the species' threat classification ($W_s$) in the final scores was tested in a sensitivity analysis (Supplementary Fig. 3).

We used one level of aggregation of agricultural threats for the calculation: Annual & Perennial Non-Timber Crops includes shifting agriculture, small-holder farming, and agro-industry farming; Wood & Pulp Plantations includes small-holder plantations, and agro-industry plantations; and Livestock Farming & Ranching includes nomadic grazing, small-holder grazing, ranching or farming, and agro-industry grazing, ranching or farming.

**Species Habitat Index (SHI)**
The SHI uses information on the size and connectivity of the species' AOH to estimate alterations in the ecological integrity of the species habitat in an area. It is a two-step approach obtained by calculating two Species Habitat Scores (SHS), the Area Size Score and the Connectivity Score, for each species in a region and then aggregating them to derive the SHI for the region[18,19].

The size of suitable habitat area in region $j$ for species $s$ ($A_{s,j}$) is given by the sum of the pixel-level suitability in region $j$:

$$A_{s,j} = \sum a \times S_{n,s} \qquad (9)$$

where $a$ is the pixel area and $S_{n,s}$ is the percentage of suitable habitat in pixel $n$ for species $s$.

The connectivity of suitable habitat area in region $j$ for species $s$ ($CS_{s,j}$) is calculated based on pixel-level presence-absence binary maps for the species in the region. A pixel with less than 1% of its area covered with habitats that are suitable for a species was coded as non-suitable (absent) in the binary map of this respective species, to reduce the influence of spurious, isolated patches[29]. For each suitable pixel, the Euclidean distance (i.e., as the crow flies) to the nearest non-suitable cell is estimated. The connectivity is then given by the average Euclidean distance of all suitable pixels from the nearest edge (i.e., GISFrag metric, ref. 42).

The Area Size Score, $AS_{s,j,k}$, and the Connectivity Score, $CS_{s,j,k}$, for species $s$, in region $j$ for a particular year $k$ (here 2021), in relation to the baseline 1 are then:

$$AS_{s,j,k} = \frac{As,j,1 - As,j,k}{As,j,1}100 \quad (9)$$

$$CS_{s,j,k} = \frac{Cs,j,1 - Cs,j,k}{Cs,j,1}100 \quad (10)$$

The mean of the Area Score and the Connectivity Score is the SHS for species $s$ in region $j$ and year $k$:

$$SHS_{s,j,k} = \frac{AS_{s,j,k} + CS_{s,j,k}}{2} \quad (11)$$

The Species Habitat Index of region $j$ in year $k$ ($SHI_{j,k}$) is then the average of the SHS for all $N$ species in the region:

$$SHI_{j,k} = \frac{\sum_{s=1}^{N} SHS_{s,j,k}}{N} \quad (12)$$

To calculate the global weighted SHI, each species' SHS was multiplied by its range fraction in location $j$ (Eq. 5) and threat level ($TL$) as a weight for threatened endemism (Eq. 6). The global weighted SHI for region $j$ in year $k$ is then:

$$SHI_{j,k} = \frac{\sum_{s=1}^{N}(TL_{s,g}ER_{s,g,j}SHS_{s,j,k})}{N} \quad (13)$$

We applied the indicators to assess three different scales of impact on biodiversity: local impact (with the cSAR), regional impact at two levels (1) ecoregion and (2) geographical mesoregions (with the cSAR and the SHI) and global impact (with the cSAR, the SHI and the STAR). When assessing local impact, location $j$ was equivalent to each 5-km-resolution pixel within the Cerrado. When assessing regional impact at the level of ecoregion and geographical mesoregions, location $j$ was equivalent to the total extent of the Cerrado ecoregion or to the area of each mesoregion within the Cerrado's extent, respectively (see Supplementary Fig. 1). Calculating local impact with SHI is also possible. However, for a large area like the Cerrado ecoregion, such calculation is very computationally intensive and was not performed in this study.

All analyses and maps were done with R Studio software (R Core Team version 4.2.2 2021)[43], with packages terra (version 1.7-71), raster (version 3.6-26), data. table (version 1.15.4), exactextractr (version 0.10.0) and caret (version 6.0-94).

### Reporting summary
Further information on research design is available in the Nature Portfolio Reporting Summary linked to this article.

## Data availability

All data on species (i.e., range distribution maps, habitat preferences, threats, etc.), land use and land cover, elevation digital models, and administrative borders supporting the findings of this study are publicly available on IUCN Red List of Threatened Species (https://www.iucnredlist.org/), Mapbiomas platform (https://brasil.mapbiomas.org/en/), Open DEM and the Brazilian Institute for Geography and Statistics' website (https://www.ibge.gov.br/en/geosciences/territorial-organization/territorial-meshes/). All data produced in the study, as well as Source Data for Table 2, Figs. 1, 2, 3, 4 and 5, and Supplementary Figs. 2 and 3 are readily available on Zenodo repository (https://doi.org/10.5281/zenodo.11352608)[44]. Source Data for generating Supplementary Fig. 1 is publicly available on Mapbiomas platform and the Brazilian Institute for Geography and Statistics' website.

## Code availability

All coding used in this study is available on Zenodo repository (https://doi.org/10.5281/zenodo.11352608)[44].

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

## Acknowledgements

G.R., T.K., M.P., and C.W. acknowledge funding from the German Federal Ministry for Economic Cooperation and Development (GRADED project, grant number GS22 E1070-0060/029). T.K. and M.P. acknowledge funding from the Belmont Forum SSCP BEDROCK project, through the German Research Foundation (DFG, grant KA 4815/2-1) and Formas (grant 2022-02563). CW acknowledges funding from the UK Research and Innovation's Global Challenges Research Fund (UKRI GCRF) Trade, Development and the Environment Hub project (grant number ES/S008160/1). G.R. and T.K. acknowledge funding from the German Federal Ministry of Education and Research (TransRegBio project, grant 031B0901A).

## Author contributions

G.R. and T.K. conceived the study. G.R. performed the analyses and developed the figures, with inputs from T.K., M.P., and C.W.; G.R. led the writing of the manuscript with inputs from T.K., M.P., and C.W.; All authors contributed to discussion of content and review.

## Funding

## Competing interests

The authors declare no competing authors.
