## [Peer Review file · Nature Communications]

Choosing fit-for-purpose biodiversity impact indicators for agriculture in the Brazilian Cerrado ecoregion

Corresponding Author: Professor U. Martin Persson

Version 0:

Reviewer comments:

Reviewer #1

(Remarks to the Author)

The study area is a region of high biological diversity, but at the same time suffering a rapid loss of natural areas to the detriment of agricultural expansion.

The analysis of these losses, as well as the evaluation of trends in the advance of new agricultural areas over remaining natural areas, are very important research topics, especially when associated with biodiversity indicators for these areas.

The findings of this study can effectively contribute to defining strategies for environmental conservation and reducing the advance of agriculture into new areas.

The methodology is adequately presented, as are the findings. As for the discussion, I expected something that goes beyond a discussion of the data used in the work. The large number (48) of mesoregions gives an idea of the diversity of environments, not only in terms of vegetation cover and land use, but also in terms of the biophysical characteristics of mesoclimate, relief and soil, for example. To what extent these aspects may or may not have contributed to making one mesoregion more susceptible to agricultural use than others, be it flat topography favoring agricultural mechanization or the availability of water for crops or animal watering.

Maybe a look at the criteria used to delimit the mesoregions of the Brazilian Cerrado (https://www.researchgate.net/publication/341089705_Caracteristicas_gerais_da_paisagem_do_bioma_Cerrado) could help in the discussion of the data and bring the findings closer to the real regions.

Furthermore, the expansion of agriculture from traditional areas to new agricultural frontiers in the south and southeast of the country has also influenced the advance into new agricultural areas, with the result that areas in the south and southeast of the Cerrado have been the scene of greater losses of biological diversity.

Finally, I made additional suggestions in the pdf file of the manuscript received regarding the figures presented, also with the aim of making it easier for readers to understand the article.

Reviewer #2

(Remarks to the Author)

This paper applied three published biodiversity impact indicators - countryside-Species Area Relationship (cSAR), Species Threat Abatement and Restoration (STAR) and Species Habitat Index (SHI)—to the Brazilian Cerrado to compare these indicators in providing impact assessment and decision support. In their calculations, the researchers used publicly available species data from the IUCN and land use data from Mapbiomas. They conclude that all three indicators show the impact of agricultural land uses on biodiversity, with cSAR and SHI assessing historical LU change impacts, whilst STAR offering potential for forward projection of impact risks. The paper is very well written throughout and easy to read. However, my main comments are:

1. This study used public data and established methodology, so what is the originality exactly? The authors can make it clearer in the introduction about the uniqueness of this work. Has such an indicator comparison been done before? What are

the known pros and cons of each of the indicators?

2. As an indicator comparison, the results seem largely as expected. But one would want such a piece in Nature Communications to provide real insights into how the potential forward-looking projections of impact risks and their value in mitigation planning can be realised. For example, it is very briefly said in the discussion that “START mapping of biodiversity risk hotspots can facilitate the identification of areas of conservation importance and shows that the abatement of agricultural expansion, especially in the eastern Cerrado and in localized regions of its transition zones with the Atlantic and Amazon forests and the Pantanal wetland, would contribute the most to avoiding global biodiversity extinctions.” I think the results and discussion on these aspects should be substantially developed for this work to be informative.

3. There need to be some critical evaluation of this work in the discussion. What are the issues involved in applying the indicators? What are the effects of data gaps and data quality? How sensitive are these indicators to data accuracy and ambiguities in the LU categories?

4. SHI results are said to be similar to cSAR but not shown?

Some minor comments:

1. Please cite the original publications where the indicators were developed.

2. Please specify the R packages used.

3. There should be a clear definition (perhaps a table) showing which LU types are considered agricultural. It occurred to me that forestry is considered agricultural use in this study, is that right? How about mixed uses within semi-natural land cover, such as agroforestry, grasslands not classified as pasture but grazed by livestock, etc.?

Version 1:

Reviewer comments:

Reviewer #2

(Remarks to the Author)

This revised manuscript has much improved clarity and insights into the results. My comments have been clearly addressed and I particularly like the now extended discussion on how these results can potentially inform complex global supply-chain activities. Just to point out that some of the new text can benefit from a round of editing check to correct minor issues. For example, the sentence on Brazil's Green Revolution is repeated.

CHOOSING FIT-FOR-PURPOSE BIODIVERSITY IMPACT INDICATORS IN AGRICULTURE		
Dear reviewers,		
we thank you for your constructive feedback and critique. We have considered all comments and made the necessary revisions to address them, which contributed to a significant improvement of our manuscript. Please see the point-by-point response below.		
On behalf of all authors, sincerely, GR		
Reviewer	Comment	Response
1	The study area is a region of high biological diversity, but at the same time suffering a rapid loss of natural areas to the detriment of agricultural expansion. The analysis of these losses, as well as the evaluation of trends in the advance of new agricultural areas over remaining natural areas, are very important research topics, especially when associated with biodiversity indicators for these areas. The findings of this study can effectively contribute to defining strategies for environmental conservation and reducing the advance of agriculture into new areas. The methodology is adequately presented, as are the findings.	Thank you for your considerations about the contributions of the study to the field.
1	As for the discussion, I expected something that goes beyond a discussion of the data used in the work. The large number (48) of mesoregions gives an idea of the diversity of environments, not only in terms of vegetation cover and land use, but also in terms of the biophysical characteristics of mesoclimate, relief and soil, for example. To what extent these aspects may or may not have contributed to making one mesoregion more susceptible to agricultural use than others, be it flat topography favoring agricultural mechanization or the availability of water for crops or animal watering. Maybe a look at the criteria used to delimit the mesoregions of the Brazilian Cerrado (https://www.researchgate.net/publication/341089705_Characteristicas_gerais_da_paisagem_do_bioma_Cerrado) could help in the discussion of the data and bring the findings closer to the real regions. Furthermore, the expansion of agriculture from traditional areas to new agricultural frontiers in the south and southeast of the country has also influenced the advance into new agricultural areas, with the result that areas in the south and southeast of the Cerrado have been the scene of greater losses of biological diversity.	Thank you for your suggestion. We reorganized our discussion to include the diversity of the social-ecological landscapes in the Cerrado also as a lens through which to think the biodiversity outcomes of our indicator comparison. We used a variety of mesoregions (e.g. south Goiás, north Mato Grosso, southeast Mato Grosso do Sul, etc.) together with the mesoregions belonging to the MATOPIBA region as examples of areas with contrasting histories of territorial occupation and biogeography and what these different backgrounds might represent for implementation of actions based on the results of the assessments.
1	Finally, I made additional suggestions in the pdf file of the manuscript received regarding the figures presented, also with the aim of making it easier for readers to understand the article	Thank you for your detailed suggestions. See point-by-point responses below.
1	I would like to see here an idea of the total area of the Brazilian Cerrado (km ² , % of South America, for example).	We included the total area of the Cerrado ecoregion in km ² . The sentence now reads: "The Cerrado is the second largest ecoregion in South America, covering around 2 million km ² and the most biodiverse savanna, holding 5% of the world's biodiversity of animals and plants, including many endemic species."
1	It would also be useful to have a scale bar accompanying each map. This gives the reader a reference, since the maps show the Brazilian Cerrado as an island. The only exception is Figure 1, which shows a map of the location of the Brazilian Cerrado in Brazil/South America, but which does not replace a scale bar. The Brazilian Cerrado is familiar to a Brazilian, not necessarily to a reader unfamiliar with Brazil and its natural regions.	We included a scale bar in all figures.
1	In Supplementary Figure 1, it would also be interesting to have an idea of area or proportion of the different LULC of the Brazilian Cerrado. Legend colors of this map could also be improved. Light colors look better on a white background.	Thank you for your suggestions. We added the cover percentages of each land use shown in the figure in the respective caption. We also changed the color scheme into lighter colors and added a scale bar.
1	25 km ² is the same as 5 km-resolution mentioned in line 88. I understood that the 5 km resolution is used in all cases,	We changed the text to avoid any confusion. The sentence now reads: "Finally, we used the cSAR to assess local impacts —risk of biodiversity loss at 5km pixel-resolution."
1	When using MapBiomass LULC data, you should inform the correspondent Collection. Since the last year of the data series is 2021, I supposed you used Collection 7 (it is the equivalent to the IUCN Red List version). Each collection of MapBiomass may apply changes not only by adding a new year to the series but also to previous years if necessary. For details, see ATBD at MapBiomass platform (https://brasil.mapbiomas.org/en)	Thank you for reminding us of this. We included the corresponding collection used in the study in the methods but also in the main text. The sentence now reads: "We used 30-m-resolution LULC maps from 2021 from the Mapbiomas platform (Collection 7.0), (...)"
1	Vilela, G.F.	We corrected the citation.
1	Should not INPE be the author here?	We corrected the citation.
2	This paper applied three published biodiversity impact indicators - countryside-Species Area Relationship (cSAR), Species Threat Abatement and Restoration (STAR) and Species Habitat Index (SHI)—to the Brazilian Cerrado to compare these indicators in providing impact assessment and decision support. In their calculations, the researchers used publicly available species data from the IUCN and land use data from Mapbiomas. They conclude that all three indicators show the impact of agricultural land uses on biodiversity, with cSAR and SHI assessing historical LU change impacts, whilst STAR offering potential for forward projection of impact risks. The paper is very well written throughout and easy to read.	We appreciate your considerations. Below we provide a point-by-point response and indicate the correspondent changes in the manuscript.
2	However, my main comments are: 1. This study used public data and established methodology, so what is the originality exactly? The authors can make it clearer in the introduction about the uniqueness of this work. Has such an indicator comparison been done before? What are the known pros and cons of each of the indicators?	Thank you for your comment. We have now stated more clearly in the introduction the uniqueness of our work. The text now reads: "In recent years, considerable effort has been made to provide end-users with guidance on the suitability of biodiversity impact indicators. Previous comparisons between indicators, however, have not used the same input data, making it difficult for users to understand why results differ and increasing the risk of drawing misleading conclusions or applying the wrong indicators. Importantly, such lack of standardised inputs obfuscates the influence of data uncertainties vs indicator choice and design. In contrast, here we use the same input data (i.e., taxa coverage, geographic extent, LU configuration) to assess how the scopes of different indicators affect results and in which ways they are useful for providing decision-makers with information on the role of agriculture as a biodiversity loss driver along supply chains."
2	2. As an indicator comparison, the results seem largely as expected. But one would want such a piece in Nature Communications to provide real insights into how the potential forward-looking projections of impact risks and their value in mitigation planning can be realised. For example, it is very briefly said in the discussion that "START mapping of biodiversity risk hotspots can facilitate the identification of areas of conservation importance and shows that the abatement of agricultural expansion, especially in the eastern Cerrado and in localized regions of its transition zones with the Atlantic and Amazon forests and the Pantanal wetland, would contribute the most to avoiding global biodiversity extinctions." I think the results and discussion on these aspects should be substantially developed for this work to be informative.	Thank you for your comment. We expanded our discussion to address how the insights being provided in our study feed back to practical actions for decisions considering the diversity of social-ecological systems within the Cerrado region and how the actors involved in such decisions relate to their different histories of territorial occupation and biophysical conditions which, in turn, also determines which indicators are more fit-for-purpose. In the restructured text, we highlight what the different insights from metrics based on comparable input sources mean for local governance, but also for different stakeholders along supply-chains, considering regions within a spectrum going from traditional agricultural areas to agricultural frontiers.

2	3. There need to be some critical evaluation of this work in the discussion. What are the issues involved in applying the indicators? What are the effects of data gaps and data quality? How sensitive are these indicators to data accuracy and ambiguities in the LU categories?	Thank you for your suggestion. We included a paragraph in the discussion to tackle data issues. The text now reads: "The advanced LULC mapping data provided by Mapbiomas for the Cerrado considerably reduces data gaps which would otherwise act as a considerable source of uncertainty for the analysis presented in this paper and—unless there are systematic biases in errors of omission and commission for given LUs or across the landscape—misclassifications are not likely to affect the quantitative results to any great extent. However, if a goal of the application of biodiversity metrics of this kind is to attribute biodiversity loss to a suite of commodities (such as might be warranted via Life Cycle Assessment activities), uncertainties associated with this type of analysis are likely to get much larger as spatial data on cultivation is only available for selected crops. Additionally, for other parts of the world where ongoing conversion of forests and other natural ecosystems is happening data availability poses a serious issue and is likely to affect the accuracy of biodiversity indicator assessments."
2	4. SHI results are said to be similar to cSAR but not shown?	We are sorry that the text caused confusion about what is to be presented in the manuscript and what is to be found in the Supplementary Information. The last paragraph of the results now reads: "The standard application of the SHI approach does not attribute loss of habitat integrity to specific LU types. For such attribution to be possible, the changes in each LU type must be tracked in the LULC maps and then proportionally allocated to the losses in habitat area size and connectivity driven by the LU transformation. By building on the original calculations for the area size component of the SHI, we explored a complementary way to use the SHI approach to attribute loss in habitat area in Cerrado mesoregions to specific LUs. This exercise can be found in Supplementary Fig. 2. The shares of the specific LU types in the total loss that resulted are similar to those found with cSAR (Fig. 3)."
2	Some minor comments: 1. Please cite the original publications where the indicators were developed.	We included the publication by Pereira and Daily (2006), where they proposed the countryside-SAR (reference 15) and the publication where GEO BON introduces the SHI, as well as the indicator metadata sheet where the methodological steps are described (references 18 and 19).
2	2. Please specify the R packages used.	We specified the R packages used and their respective versions.
2	3. There should be a clear definition (perhaps a table) showing which LU types are considered agricultural. It occurred to me that forestry is considered agricultural use in this study, is that right? How about mixed uses within semi-natural land cover, such as agroforestry, grasslands not classified as pasture but grazed by livestock, etc.?	Thank you for pointing this out. We are sorry that this caused confusion. We included a description of what LU types are considered agricultural in the main text, expanded the description of LUs in the methods and added a table in the Supplementary Information with the descriptions from Mapbiomas for the LULC classes. The sentence in the main text now reads: "We used (...) land use and land cover (LULC) maps from Mapbiomas (Collection 7.0). We considered agricultural LU all classes with specific crops, cropland, pasture (or a mix of the latter two), as well as monocultural tree plantations (Supplementary Table 1). LUs related to extractivism or mixed uses within semi-natural land cover were not included in this collection of LULC maps."

Point-by-point response to reviewer comments

Reviewer comment:

Response:

This revised manuscript has much improved clarity and insights into the results. My comments have been clearly addressed and I particularly like the now extended discussion on how these results can potentially inform complex global supply-chain activities. Just to point out that some of the new text can benefit from a round of editing check to correct minor issues. For example, the sentence on Brazil's Green Revolution is repeated.

Thank you for the support. We are glad that we were able to address your concerns and improve the quality of our manuscript. We removed the repeated sentence and double-checked the whole text for any other typos.